# Pre-Reconstruction Processing with the Cycle-Consist Generative Adversarial Network Combined with Attention Gate to Improve Image Quality in Digital Breast Tomosynthesis

**DOI:** 10.3390/diagnostics14171957

**Published:** 2024-09-04

**Authors:** Tsutomu Gomi, Kotomi Ishihara, Satoko Yamada, Yukio Koibuchi

**Affiliations:** 1School of Allied Health Sciences, Kitasato University, Sagamihara 252-0373, Kanagawa, Japan; 2Department of Radiology, NHO Takasaki General Medical Center, Takasaki 370-0829, Gunma, Japan; 3Department of Breast and Endocrine Surgery, NHO Takasaki General Medical Center, Takasaki 370-0829, Gunma, Japan

**Keywords:** breast tomosynthesis, attention gate, cycle-consistent generative adversarial networks, image quality improvement

## Abstract

The current study proposed and evaluated “residual squeeze and excitation attention gate” (rSEAG), a novel network that can improve image quality by reducing distortion attributed to artifacts. This method was established by modifying the Cycle Generative Adversarial Network (cycleGAN)-based generator network using projection data for pre-reconstruction processing in digital breast tomosynthesis. Residual squeeze and excitation were installed in the bridge of the generator network, and the attention gate was installed in the skip connection between the encoder and decoder. Based on the radiation dose index (exposure index and division index) incident on the detector, the cases approved by the ethics committee and used for the study were classified as reference (675 projection images) and object (675 projection images). For the cases, unsupervised data containing a mixture of cases with and without masses were used. The cases were trained using cycleGAN with rSEAG and the conventional networks (ResUNet and U-Net). For testing, predictive processing was performed on cases (60 projection images) that were not used for learning. Images were generated using filtered backprojection reconstruction (kernel: Ramachandran and Lakshminarayanan) from projection data for testing data and without pre-reconstruction processing data (evaluation: in-focus plane). The distortion was evaluated using perception-based image quality evaluation (PIQE) analysis, texture analysis (feature: “Homogeneity” and “Contrast”), and a statistical model with a Gumbel distribution. PIQE has a low rSEAG value. Texture analysis showed that rSEAG and a network without cycleGAN were similar in terms of the “Contrast” feature. In dense breasts, ResUNet had the lowest “Contrast” feature and U-Net had differences between cases. The maximal variations in the Gumbel plot, rSEAG reduced the high-frequency ripple artifacts. In this study, rSEAG could improve distortion and reduce ripple artifacts.

## 1. Introduction

Screening mammography is associated with not only decreased breast cancer mortality rates but also high breast cancer detection rates at an early stage. In particular, diagnosing breast cancer using this method may decrease the need for more radical surgical treatment and chemotherapy [1]. Mammography is primarily used for specific and general screening and breast cancer diagnosis in the early stages. Digital breast tomosynthesis (DBT) is a promising tool in both areas.

The disadvantages of mammography are associated with the composite overlapping of fibroglandular tissues. In cancer diagnosis, dense fibroglandular tissues may cause a blurred visualization of a malignancy. Thus, the diagnostic value of the method decreases. In theory, DBT can compensate for the indicated limitations of mammography. In contrast to conventional mammography, which conveys radiography using tissue volume and builds an image from the absorbed X-rays from all tissues, DBT concentrates on a thin slice of tissue that is of interest while obscuring all other tissues.

Recent studies have shown increasing interest in deep learning models (DLMs). Their application in medical image reproduction is directly associated with that of basic models, such as generative adversarial networks (GANs). GANs bypass the uncompliant probabilistic assumptions using a standardized network for data distribution. This sampling network is used in an adversarial fashion, in which the data samples that are generated are as real as possible and a discriminator distinguishes real samples from the ones that were generated.

DBT studies can clearly identify masses and have an enhanced image quality in processes that use DLMs [2,3,4,5,6,7]. Regarding image processing with conditional GAN (cGAN or supervised image-to-image translation [pix2pix]) [8], pix2pix, which aligns the object image with the reference with an adversarial network using a generator and discriminator, was beneficial for contrast structure (CS) improvement and noise reduction.

Deep learning was applied directly to image pre-reconstruction. First, a reconstruction algorithm (exact filtered backprojection: FBP or iterative) was developed to transform raw data (projection data) into a medical image. Considering the limitations of the reconstruction algorithm, DLM that used a low-quality medical image as the input and released the improved image as the output was applied. For example, the image had a higher resolution or a lower level of noise or fewer artifacts. This was referred to as the pre-reconstruction image-domain approach, which worked with the reconstructed image, not including the reconstruction pipeline. Thus, existing image-to-image deep neural networks (pix2pix), such as U-Net [9], could be applied to address this issue.

The conventional approach for image processing and building is associated with challenges that limit the identification of masses and the accurate preservation of their normal structure [10,11,12,13,14,15,16,17]. As the level of noise in low-dose imaging increases, the detection of masses and the preservation of normal structures (e.g., structural distortion, image sharpness, and artifacts) occur simultaneously.

The image quality and radiation dosage of several DBT algorithms have been mathematically examined [18,19]. Previous studies have shown that iterative reconstruction may substantially lower the level of noise and radiation exposure. However, their analyses were limited, and they focused on a comparative assessment of existing methods. Liu et al. used a controlled image processing technique to train a neural network using a linear-output-layer backpropagation algorithm. By applying and evaluating low-dose images in a limited number of clinical cases, the proposed method was found to be a promising technology for low-dose reduction [20]. Recently, Gao et al. revealed that diminishing noise in a deep convolutional neural network with adversarial training is helpful in improving microcalcification (MC) CS in DBT using in silico data as a learning set [21]. Gomi et al. compared DBT images generated by applying pix2pix with the low-dose pre-reconstruction domain and conventional image processing methods. Results showed that pix2pix was useful in upgrading the quality of low-dose images. Nonetheless, the reconstruction algorithm was only FBP, and the evaluation cross-section was an in-focus plane with limited information [22]. Among the studies that have used deep learning, only a few performed a quantitative assessment of image quality refinements, reduction of the MC dose, and identification of masses in diverse circumstances using automatic exposure control (AEC) as the comparative dose because of variations in breast thickness. In particular, because it can decrease the radiation dose and upgrade image quality, this logic (i.e., an image-to-image translation process, which allows the application of low-dose images to reference-dose images) can be used in DBT to upgrade image quality degradation under low doses.

A previous study investigated supervised learning approaches for DBT image reconstruction [22]. However, it is impractical to use supervised models because of the absence of paired training data. For example, in noise reduction, artifacts and noise are derived from low-dose conditions. Therefore, capturing a pair of images is not appropriate, as one is low-dosed and the other is not under a low-dose condition, which is not supposed to yield practical results. In DBT, images are inherently noisy because of the diminished number of photons emitted by the flat-panel detector. Therefore, there are no available clean ground-truth images. Unsupervised learning approaches are used to address these scenarios. Considering low-dose image data, the models that could learn DBT image reconstruction without high-quality ground-truth data were evaluated. GANs could produce data that are statistically more comparable with the training samples. Considering image reconstruction, GANs can generate high-quality images without supervision. This is an interesting property when combined with a GAN-based method using residual neural network (ResNet) [23] techniques, such as cycle-consistent GAN (cycleGAN) [24], because it provides opportunities for teaching DBT image reconstruction from unpaired training samples.

The use of ResUNet, which is an improved version of U-Net, is useful for improving image identification accuracy and reducing artifacts [25]. ResUNet is based on U-Net, with a residual mechanism added to the encoder and decoder. Therefore, compared to U-Net, it is possible to improve the identification accuracy of feature maps in each layer.

Based on recent studies, attention mechanisms have been adopted in deep learning to improve image recognition and classification accuracy [26,27,28]. Attention calculates the dot product of the vector corresponding to the original feature map. Then, with respect to the generated coefficients and the feature map (part) of the next output layer, coefficients are made larger for areas with a strong association and smaller for areas with a weak association. Using this attention mechanism, a feature map that emphasizes the target region of interest can be generated. In a conventional convolutional neural network (CNN), learning ranges from simple patterns between pixels to complex patterns for input images. However, the attention mechanism realizes the learning of complex relationships from simple relationships among multiple feature maps. Therefore, by incorporating this attention mechanism into a part of the generator network, the identification accuracy of the feature map of each layer can improve and contribute to obtaining a better final target image. In addition, another study has reported that incorporating an additive attention gate (AG), which is a developed attention mechanism between U-Net system skip connections, is useful [26].

In this study, by inserting residual, squeeze, and excitation (SE) [29] into the bridge, the structural extraction of normal tissues and diseased areas improved. Incorporating an AG between the skip connections further improved the accuracy of separating normal and mass structures.

CycleGAN was selected as the base system in constructing a network for improvement. This is because the training data were unpaired samples, and a network comprising two pairs of generators and discriminators was required to improve the accuracy of the extraction of minute masses required for breast diagnosis. In addition, an encoder and a decoder structure, which allowed the flexible insertion of an AG into the process related to skip connection and SE in the bridge was preferred for the configuration of the generator. ResUNet, an optimal network based on U-Net, has a residual network configuration that improves the network’s ability to discrimination between features in each convolution layer, was consistent with this condition.

Based on this logic, to improve image quality and reduce distortion attributed to artifacts, we used ResUNet in a cycleGAN-based generator network to perform pre-reconstruction processing for projection data, inserted residual and SE in the bridge, and skipped the connection from encoder to decoder. Thus, a novel residual squeeze and excitation attention gate (rSEAG) that incorporates AG in between was proposed. The image reconstruction algorithm used FBP to calculate the exact solution of projection data.

To validate the usefulness of the proposed rSEAG, the conventional networks U-Net and ResUNet were compared. Moreover, the use of perception-based non-reference-type comprehensive image quality evaluation (PIQE) was examined [30]. Subsequently, basic image features such as “Homogeneity” and “Contrast” were calculated via texture analysis [31]. Finally, the evaluation was performed by the streak artifacts of the high-frequency components using a statistical model with a Gumbel distribution [32].

## 2. Materials and Methods

### 2.1. DBT

A DBT system (Selenia Dimensions; Hologic Inc., Bedford, MA, USA), with an X-ray tube with a 0.3-mm focal spot (tube target, W; filtration, 0.7-mm aluminum equivalent) and digital flat-panel amorphous selenium detector was used. The parameters of all DBT procedures were as follows: total acquisition time: 3.7 s and acquisition angle: 15°. The projection images were sampled during a single tomographic pass (projections: 15, matrix: 2560 × 4096). A cropped image (matrix: 1024 × 2048) with 32 bits (single-precision floating numbers) per image was used.

### 2.2. Cases and Datasets

This study was approved the institutional review board (TGMC2023-009, NHO Takasaki General Medical Center, approved 6 June 2023). The recruitment of participants for this study lasted from 7 June 2023 to 17 August 2023 (Table 1). Further, it used existing information without details on invasiveness or intervention. Due to some difficulties, written or oral informed consent was not obtained from the participants. Information about the research was posted in the hospital (NHO Takasaki General Medical Center) or the hospital’s website (opt-out) and was disclosed to the participants. Moreover, the participants were informed about the option to drop out of the study. All of the patients underwent DBT examinations. Projection images comprised various patterns that included either or both the imaging direction and examination area (left and right). The datasets collected using AEC at reference radiation doses were considered as references. Meanwhile, the datasets collected using AEC (reference: 45 [with and without cases], object: 45 [with and without cases]) under low radiation doses were considered as objects. Using data collected under low-radiation dose conditions as an object, images with degraded image quality caused by factors including artifacts and noise could be improved. Reference and object were classified based on the exposure index (EI) and deviation index (DI) [33,34,35], which are indicators of the radiation dose incident on the detector (testing data, case_1: 50-year-old patient with invasive ductal carcinoma (IDC) with heterogeneous dense breasts [left cranio-caudal, LCC]; case_2: 50-year-old patient with IDC with heterogeneous dense breasts [left medio-lateral oblique, LMLO]; case_3: 48-year-old patient with IDC [RCC]; case_4: 48-year-old patient with IDC [RMLO]). Case_1 and case_2, case_3 and case_4 are the same patients. The testing dataset, which is the target of prediction processing in the training data, was picked up from the object and separated from the training data. Regarding the data used for learning, 675 projection images were used as reference and object, and 60 sets were used for testing. In total, 128 images were set as patch images (128 × 128 pixels) for each image. Therefore, the total sample images were 86,400 each for reference and object. Each image related to the input images was randomly selected and flipped horizontally.

### 2.3. cycleGAN

In the unpaired learning of DBT image building, we assumed that both the reference and object DBT images were disposable for training. However, a pairwise correlation was not observed between the reference and object DBT images. Considering an object image from the training set, the ground-truth reference image was not identified. The goal was to develop a deep neural network that converts an object image into a reference under these unpaired parameters and without direct control. ResNet was used for the generator structure, PatchGAN [8] for the discriminator structure, and an optimization algorithm for Adam [36]. Appendix A (Table A1 and Table A2) presents the detailed network architecture and hyperparameters of the building components of cycleGAN (Table 2) and each generator network.

### 2.4. AG

Introducing an AG to ResUNet skip connections suppresses irrelevant feature representations in the processing system during encoding network sampling. This allows the model to retain more relevant spatial information, thereby reducing computational cost. This can lead to reduced artifacts. Figure 1 shows the AG module. The principle is to suppress the features of the irrelevant background area through the channel signal based on the spatial information of the image [28]. In particular, channel signals were used to aggregate the multilevel features of skip connections to improve the performance of feature extraction to extract spatial information. AG allows you to obtain gate coefficients even in linear and complex systems, thereby achieving an efficient context modeling. Equations (1) and (2) present the AG:(1)f=ηReLuWyy+Wωω
(2)β=Sigfy,ω;TriLi
where ηx is an element-wise nonlinearity (rectified linear unit, ReLu) and Sigx is a normalization function (sigmoid). The input feature is y (from concatenate, y∈ℝH×W×C
*H*: height, *W*: width, and *C*: channel) and ω (from bridge & decoder, ω∈ℝH×W×C). TriLi, which is the grid resampling of attention coefficients, is performed using trilinear interpolation with linear transformations (Wy, Wω) (Figure 1).

### 2.5. rSEAG

The novel feature of rSEAG is that it incorporates a cycleGAN-based generator network with a residual network, SE network, and an AG based on ResUNet (Figure 2). The key to the SE network is to apply weights adaptively, not to output each channel of the convolutional layer equally. Therefore, the input of the SE block can inherently introduce conditioning dynamics to improve the ability to identify features (Figure 3). The configuration combines with the SE network to identify the local features of the images. AG, which is an image recognition model that extracts features from the whole image, was utilized to improve the accuracy of identification.

### 2.6. Normalization of the Image Data

Normalization was applied to all analyzed and evaluated images used in this study. The two-dimensional image data were mapped to a range of values (0 or 1), where 0 represents air and 1 displays the mean pixel values of the phantom on a pixel-by-pixel basis using the following formula:(3)$$I_{temp(i,j)}=\frac{I_{org(i,j)}-{\bar{I}}_{org(i,j)}}{\sigma }$$
(4)$$I_{norm(i,j)}=1.0+\left[\frac{I_{temp(i,j)}-\min I_{temp(i,j)}}{\max I_{temp(i,j)}-\min I_{temp(i,j)}}\right]$$
where $I_{norm(i,j)}$ is the normalized image, $I_{temp(i,j)}$ is the original image, ${\bar{I}}_{org(i,j)}$ is the mean value of $I_{temp(i,j)}$, and σ is the standard deviation of $I_{temp(i,j)}$.

### 2.7. Evaluation of Optimization Parameters Foe Epochs

The optimization epochs in the cycleGAN network were assessed based on the mean square error (MSE) [37] for each projection image. After estimating the MSE of the training network, the optimal epoch was selected as the lowest MSE. The lowest MSE and the number of epochs were selected as the optimum parameters.

### 2.8. Evaluation of the Image Quality

The DBT system-obtained reconstructed projection data were used for quality evaluation (FBP kernel, Ramachandran, and Lakshminarayanan). MATLAB Ver. 23.2.0.2485118 R2023b (MathWorks, Natick, MA, USA) was used to receive and process all images (using a custom script for the MATLAB environment). The PIQE, texture analysis (gray level co-occurrence matrix [GLCM]), statistical model with a Gumbel distribution, and in-focus plane were calculated to assess the contributions of each processing method.

### 2.9. PIQE

PIQE is used to calculate a non-reference image quality score, with lower scores indicating a better perceptual quality. In recent medical imaging, it is also used as an evaluation method to improve the image quality of mammography and magnetic resonance imaging [38,39,40]. In this study, since there was no reference image, it was utilized to evaluate image quality improvement. Moreover, it was used to calculate the mean subtracted contrast normalized (MSCN) coefficients for each pixel in the image, and to estimate the distortion based on the variance of the MSCN coefficients in blocks (16 rows and 16 columns). A threshold condition (10% of the MSCN coefficient) was applied for classification (distorted blocks with artifacts, distorted blocks with noise, and undistorted blocks). Finally, the scores of the distorted blocks were calculated as an average value. Distortion caused by artifacts and the noise of each block was evaluated based on the PIQE score.

### 2.10. Texture Analysis (GLCM)

In the texture analysis, other related evaluations were performed using PIQE, which uses a non-reference image score to evaluate distortion. Factors that contribute to distortion were related by noise and pixel-to-pixel variation. Therefore, in terms of various texture analysis items for evaluation, “Homogeneity” was selected for noise and “Contrast” for image-to-image variation. Texture analysis refers to the density distribution in which the fine patterns of light and shade are uniformly distributed. GLCM is defined as a matrix representing the relationship (8 neighborhoods) with adjacent pixels (Equation (5)).
(5)$$T_{img}(i,j)=\frac{O_{img}}{\sum_{i,j}O_{img}(i,j)}$$
where $T_{img}\left(i,j\right)$ is the GLCM map, $O_{img}$ is the image to be analyzed, *i* is the number of neighbors with the center pixel value, and *j* is the neighbor pixel value.

The number of gray levels used for analysis was set to 256, and the features used for evaluation were “Homogeneity” and “Contrast”. “Homogeneity” returns a value that indicates how close the distribution of GLCM elements is to the GLCM diagonal. If the linearity is higher, the noise characteristic was lower. “Homogeneity” is defined by Equation (6).
(6)$$Homogeneity=\sum_{i,j}\frac{T_{img}\left(i,j\right)}{1+\left|i-j\right|}$$

“Contrast” returns a measure of the intensity CS between a pixel and its neighbors for the whole image. “Contrast” is defined by Equation (7).
(7)$$Contrast={\sum_{i,j}\left|i-j\right|}^{2}T_{img}\left(i,j\right)$$

### 2.11. Statistical Model with a Gumbel Distribution

Based on the Gumbel distribution, the streak artifacts of the high-frequency components found in cross-sectional images could be quantitatively analyzed [32,41]. A rectangular window with a width of 30 pixels and a length of 24 pixels (X-ray sweep direction) was placed on each in-focus plane image, nearly perpendicular to multiple streak artifacts. The region of interest was set around the ripple artifact, which occurred at the breast periphery. The parallel-line profiles of the pixel values at 1-pixel intervals resulted in a total of 29 parallel-line pixel-value profiles (each sampling size: 29). The pixel-value profiles were graphed, and the maximal variations between adjacent pixel values were determined and analyzed based on the Gumbel distribution. The cumulative probability function was measured using the symmetry rank method with order statistics: cumulative probability $Q(x_{\left(\partial \right)})=\frac{\left(\partial -0.5\right)}{l},\left(\mathrm{for}\partial =1,\ldots l\right)$, where l is the sampling size. Based on the generated probability diagram, the maximal difference between adjacent pixels showed linearity with a cumulative probability. Thus, the Gumbel characteristic could be calculated based on the maximal difference between neighboring pixels. Finally, the Pearson’s correlation coefficient was determined and examined to evaluate linearity (probability [*p*] values < 0.01) using the Statistical Package for the Social Sciences software for Windows (version 24.0; IBM Corp., Armonk, NY, USA).

## 3. Results

### 3.1. Optimization Parameters for Epochs

Considering the findings of the optimization verification, each training network image was created by establishing optimization epochs for cycleGAN (rSEAG: case_1 35, case_2 35, case_3 34, and case_4 34; ResUNet: case_1 10, case_2 10, case_3 12, and case_4 11; and U-Net: case_1 29, case_2 29, case_3 31, and case_4 30). Training was performed using the NVIDIA RTX A5000 graphic processing unit (memory: 24 GB). The overall time periods required for rSEAG, ResUNet, and U-Net for 300 epochs were 194.01, 52.61, and 109.16 h, respectively (Figure 4, Figure 5 and Figure 6).

### 3.2. Image Quality

Figure 7, Figure 8, Figure 9 and Figure 10 show the comparison results for each DBT image reconstructed by a network without cycleGAN and each tested network. In the dense breasts type, compared with networks without cycleGAN and conventional networks, rSEAG resulted in increased density in the mass area and improved CS by emphasizing increasing density. The CS improvement generated by emphasizing associations with increased density, as observed in rSEAG, was accentuated in the area centered around the mass area, rather than in normal tissues. In the non-dense breasts type, there was no clear difference in terms of CS in the mass area between networks without cycleGAN and each tested network. ResUNet had increased artifacts (undershoot-like artifacts along the X-ray sweep direction) occurring at the breast contour and the mass margin. In the case of U-Net, the density in the mass area reached saturation in the dense breasts type, and the CS decreased.

Figure 11 depicts the comparison results of PIQE for each case using a network without cycleGAN and each tested network. Compared with networks without cycleGAN, rSEAG had the lowest value among the conventional networks on all but case_1. Compared with networks without cycleGAN and rSEAG, rSEAG showed that the non-dense breasts type had a lower value. RSEAG reduced the distortion caused by artifacts and noise compared with conventional networks and networks without cycleGAN, excluding dense type cases (case_1 of network without cycleGAN and U-Net) and one non-dense type (case_4 of ResUNet).

Figure 12 presents the comparison of “Homogeneity” and “Contrast” for each case using a network without cycleGAN and each tested network. In the non-dense breasts type, the network without cycleGAN and each tested network were similar in terms of “Homogeneity”. Moreover, there was only a slight difference in the noise level. In the dense breasts type, ResUNet had the highest value among all networks, resulting in an image with a low level of noise. Among all networks, ResUNet had the lowest value, particularly in dense breast types of “Contrast”. In addition, the “Contrast” value of U-Net significantly differed among dense breasts type cases.

Figure 13 presents the Gumbel plot of the associations between the maximal variations ± standard error and predicted cumulative probabilities (case_1: without cycleGAN = 0.0490 ± 0.0030, rSEAG = 0.0248 ± 0.0016, ResUNet = 0.0632 ± 0.0033, and U-Net = 0.0277 ± 0.0013; case_2: without cycleGAN = 0.0507 ± 0.0026, rSEAG = 0.0406 ± 0.0022, ResUNet = 0.0777 ± 0.0044, and U-Net = 0.0365 ± 0.0017; case_3: without cycleGAN = 0.0054 ± 0.0004, rSEAG = 0.0056 ± 0.0008, ResUNet = 0.0085 ± 0.0006, and U-Net = 0.0038 ± 0.0002; case_4: without cycleGAN = 0.0292 ± 0.0024, rSEAG = 0.0237 ± 0.0019, ResUNet = 0.0250 ± 0.0020, U-Net = 0.0248 ± 0.0023). Here, the largest variations were linearly distributed (case_1: without cycleGAN, r = 0.990 [*p* < 0.01]; rSEAG, r = 0.990 [*p* < 0.01]; ResUNet, r = 0.963 [*p* < 0.01]; U-Net, r = 0.961 [*p* < 0.01]; case_2: without cycleGAN, r = 0.963 [*p* < 0.01]; rSEAG, r = 0.965 [*p* < 0.01]; ResUNet, r = 0.979 [*p* < 0.01]; U-Net, r = 0.933 [*p* < 0.01]; case_3: without cycleGAN, r = 0.923 [*p* < 0.01]; rSEAG, r = 0.847 [*p* < 0.01]; ResUNet, r = 0.977 [*p* < 0.01]; U-Net, r = 0.943 [*p* < 0.01]; case_4: without cycleGAN, r = 0.960 [*p* < 0.01]; rSEAG, r = 0.935 [*p* < 0.01]; ResUNet, r = 0.983 [*p* < 0.01]; U-Net, r = 0.953 [*p* < 0.01]). These observations validated the use of a Gumbel distribution as an acceptable statistical model for defining the largest variations in the nearby pixel-value profiles. In addition, based on an analysis of the maximal variations in the Gumbel plot, rSEAG had the lowest high-frequency ripple artifacts for case_1, case_2, and case_3. The ResUNet exhibited the highest high-frequency ripple artifacts. However, the distribution differed in other networks.

## 4. Discussion

CycleGAN pre-reconstruction processing with rSEAG can improve the quality of DBT images under unsupervised clinical case conditions. In this study, the in-focus plane of cycleGAN pre-reconstruction processing with rSEAG improved image quality and reduced architectural distortion in the whole mass image. Further, cycleGAN pre-reconstruction processing with rSEAG could have contributed to ripple artifact reduction. Thus, this method could be a promising novel strategy for improving the separation accuracy between normal tissues and mass areas, unlike images subjected to conventional network architectures. Moreover, the flexibility of cycleGAN pre-reconstruction processing with rSEAG is associated with a better usability.

Based on the comparative study results, rSEAG could provide images featuring an integrated image quality balance between the in-focus plane in AEC (Figure 11 and Figure 12). The ability to provide accurate image information from both in-focus plane aspects under reference-dose imaging conditions can enhance the diagnostic value of the breast area in the future. Therefore, rSEAG allows us to proceed in a direction in which, unlike conventional filtering-based image processing, unknown degraded images can be restored to a state with standard image quality by utilizing existing learning data. We believe that the method presented in our study is among the measures that could improve image quality in DBT with reference to integrated image quality, though we acknowledge that our results are experimental and based on a limited number of cases.

There are several concerns regarding the use of DBT. For example, it is associated with increased radiation exposure and the potential risk of radiation-related malignancy. DBT vendors have used various approaches to address this issue. These approaches reduce the additional dose by making the examination at a single view. Considering that the DBT dose is twice that of the usual radiation exposure for mammography, performing a single DBT can decrease the cumulative dose. In terms of accuracy, a few studies have shown that single-view DBT is least compatible with two-view mammography [42]. The use of different angular dose profiles produces high-quality projections. Nishikawa et al. revealed that a high-dose central projection can improve the speed and accuracy of MC detection while maintaining high mass detection rates in DBT reconstruction [43]. However, other studies have shown a similar [44] or inferior performance [45] for MC detection (or specks simulating MCs) using this projection. Therefore, further investigations should be performed to determine whether these projections have clinical benefits.

CNN does not require input from hidden layers obtained from a specific processing layer. Thus, calculations can be parallelized. However, the association between tokens (vectors that compress the information of the whole image) located at distant locations is challenging to identify. Attention mechanisms have been proposed to address this issue. The similarity measure of the attention mechanism is determined by the inner product of vectors and the weighted sum using the inner product as a coefficient. By creating a processing system that focuses on areas of interest, such as mass areas, using a network that combines CNN and attention, such as the proposed rSEAG, the accuracy of identifying normal from mass tissues can improve. Hence, a system that reduces distortion caused by artifacts in the mass area can be useful.

Moreover, inserting an AG between the skip connections of ResUNet, which is the basis of this generator network, could improve image quality (especially by removing distortion and reducing artifacts caused by ripples). By concatenating encoder and decoder information, the features required for updating increase, and they are effective in improving identification accuracy. The effect of improving identification accuracy can be synergized by incorporating an AG with the utilization of the rich feature values that exist between skip connections in each layer. Instead of outputting each channel of the convolutional layer equally, addition of an SE to the generator bridge could cause the input conditioned by dynamics due to adaptive weighting, which improves the discrimination ability of the features. While the AG calculates the similarity between multiple features with improved discrimination ability via SE, the attention weight, which is a coefficient that indicates the association of the similarity with improved accuracy by the dot product and weighted sum, could also be calculated. This attention weight allows learning to focus on the mass region that deserves attention in the image. Therefore, rSEAG, which combines SE and AG, has reduced the structural distortion seen in normal and mass images and reduced ripple artifacts.

CycleGAN defines cycle consistency and adversarial losses. The cycle consistency loss is based on the difference between the original image and the original image transformed to another domain and then retransformed to the original domain. Meanwhile, the adversarial loss ensures that the image is real. Training an image-to-image transform framework is often challenging, because it requires perfectly matched images. These issues can be solved by CycleGAN by allowing transformations between domains that are not perfectly matched. Therefore, unlike conventional processing for correcting differences between images, an image that approximates a reference image with high precision and efficiency can be generated.

In FBP, the measurement process is characterized by the Radon transform [46]. Moreover, the system’s analytical equation is solved before discretization of the solution. However, only approximate solutions are presented for incomplete data, and efforts have been made to identify appropriate approximations. In cycleGAN pre-reconstruction processing with FBP, effective CS preservation enabled the derivation of an appropriate approximate solution at the projection data level, thereby addressing the analytical solution for the inverse problem for DBT.

In medical images, the types, distribution, and density of breast glandular tissues and the corresponding overlaps of glandular density were inconsistent, which caused issues in the model optimization. Overcoming the effects of breast gland thickness and breast overlap owing to the use of more varied cases is a future challenge. In the next evaluation stage, to overcome these issues, learning and evaluation using clinical cases acquired at the projection data level can be used. For example, the use of samples with high and low noise levels at the projection data level was proposed. In this study, a training dataset that did not significantly differ between populations was used. A good performance was achieved on the test data because learning was performed in a state where there was no gap in the main distribution between the two domains. Further studies using more clinical data should be performed to better understand the association between learning and performance. Nevertheless, no studies have focused on improving the reproducibility or performance of unsupervised GANs. Therefore, studies that focus on evaluating the reproducibility and performance of unsupervised GANs are interesting.

This study had some limitations. One evaluation had an increased variation in the dataset used for testing. Therefore, it is important to evaluate cases of MCs and benign and malignant masses in patients with various breast types.

## 5. Conclusions

In this study, based on the comprehensive image quality evaluation in the in-focus plane, cycleGAN pre-reconstruction processing with rSEAG can improve distortion and reduce ripple artifacts.

## Figures and Tables

**Figure 1 diagnostics-14-01957-f001:**
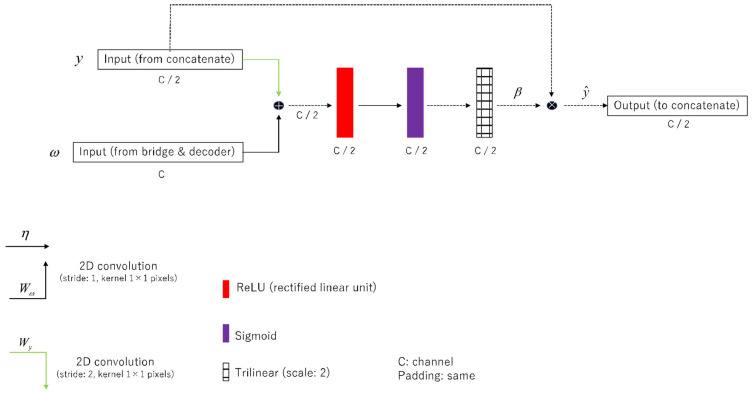
Network architectures for an additive attention gate (AG). AG diagram of the rSEAG network elements.

**Figure 2 diagnostics-14-01957-f002:**
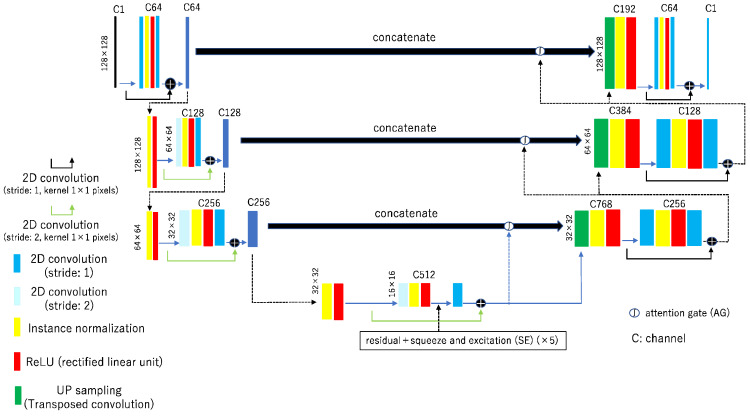
Network architectures for residual squeeze and excitation attention gate network (rSEAG). Concept of the rSEAG for digital breast tomosynthesis.

**Figure 3 diagnostics-14-01957-f003:**
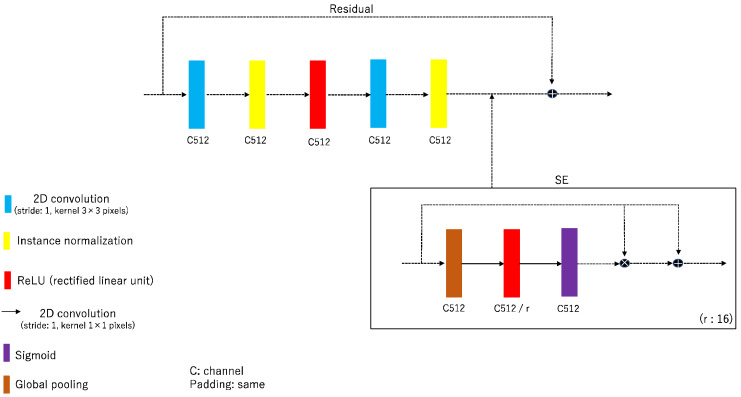
Network architectures for residual and squeeze and excitation (SE). Residual and SE diagram of the rSEAG network elements.

**Figure 4 diagnostics-14-01957-f004:**
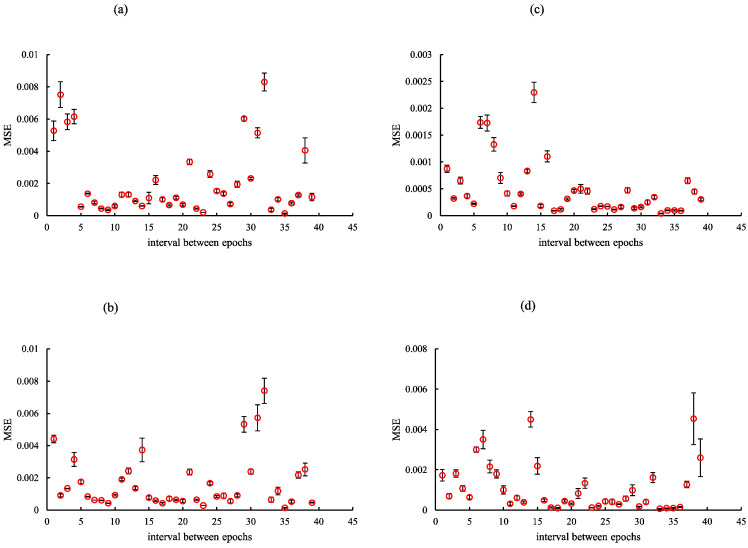
Optimization results for parameter (epochs) determination for cycle-consistent generative adversarial network pre-reconstruction processing (generator network, residual squeeze and excitation attention gate network [rSEAG]) of different testing cases. (**a**) case_1, (**b**) case_2, (**c**) case_3, and (**d**) case_4. The horizontal axis shows the interval between epochs.

**Figure 5 diagnostics-14-01957-f005:**
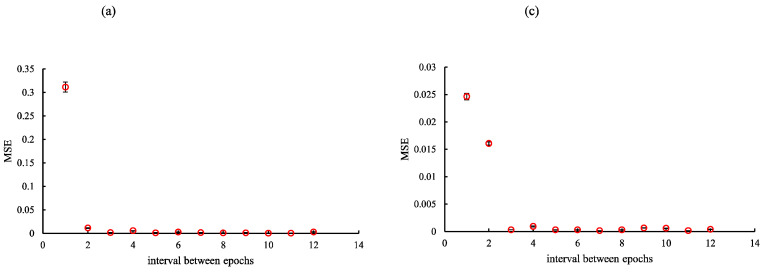
Optimization results for parameter (epoch) determination for cycle-consistent generative adversarial network pre-reconstruction processing (generator network, ResUNet) of different cases. (**a**) case_1, (**b**) case_2, (**c**) case_3, and (**d**) case_4. The horizontal axis shows the interval between epochs.

**Figure 6 diagnostics-14-01957-f006:**
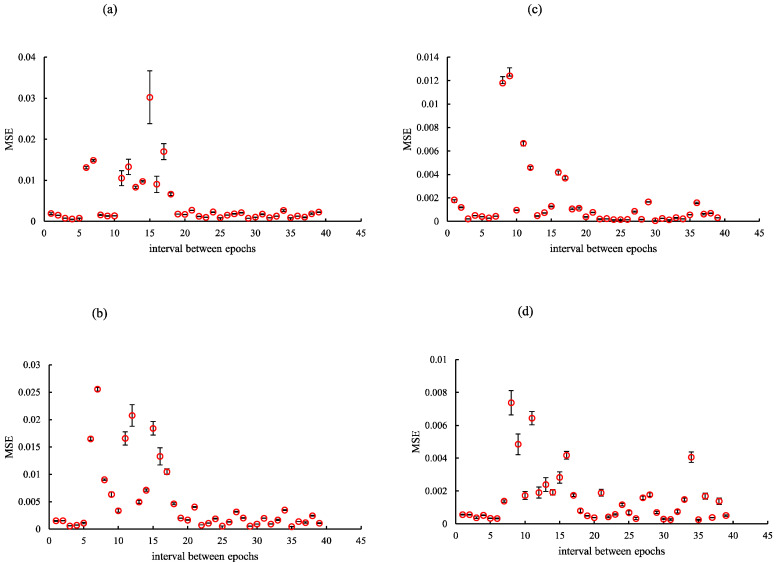
Optimization results for parameter (epoch) determination for cycle-consistent generative adversarial network pre-reconstruction processing (generator network, U-Net) of different cases. (**a**) case_1, (**b**) case_2, (**c**) case_3, and (**d**) case_4. The horizontal axis shows the interval between epochs.

**Figure 7 diagnostics-14-01957-f007:**
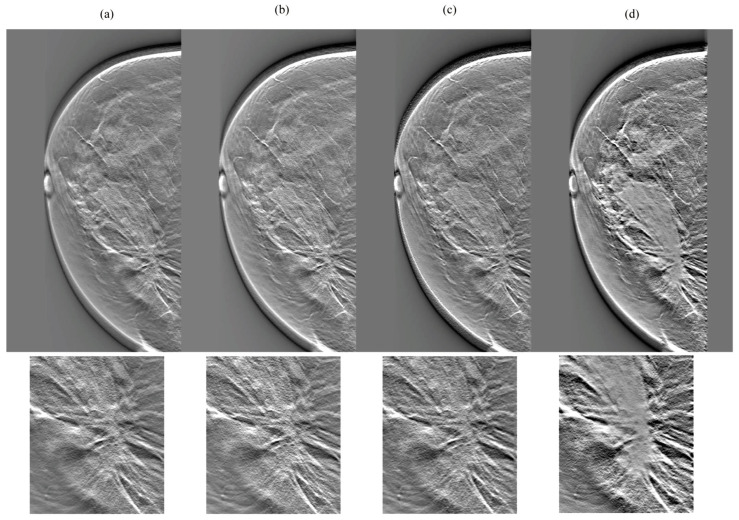
Comparisons between cycle-consistent generative adversarial network (cycleGAN) pre-reconstruction processing with and without relative generator networks (network without cycleGAN, residual squeeze and excitation attention gate network [rSEAG], ResUNet, and U-Net) hole and zoomed images in the in-focus plane (case_1). The all-display window range is 0–0.02. (**a**) Without cycleGAN, (**b**) rSEAG, (**c**) resUNet, and (**d**) U-Net.

**Figure 8 diagnostics-14-01957-f008:**
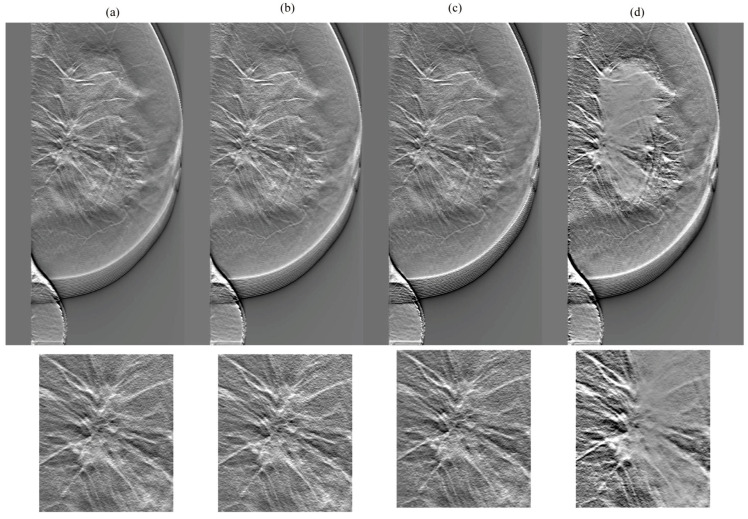
Comparisons between cycle-consistent generative adversarial network (cycleGAN) pre-reconstruction processing with and without relative generator networks (network without cycleGAN, residual squeeze and excitation attention gate network [rSEAG], ResUNet, and U-Net) hole and zoomed images in the in-focus plane (case_2). The all-display window range is 0–0.02. (**a**) Without cycleGAN, (**b**) rSEAG, (**c**) resUNet, and (**d**) U-Net.

**Figure 9 diagnostics-14-01957-f009:**
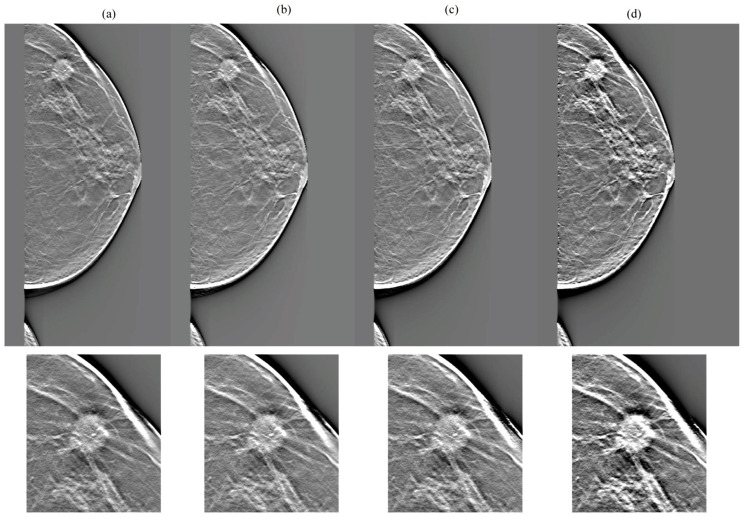
Comparisons between cycle-consistent generative adversarial network (cycleGAN) pre-reconstruction processing with and without relative generator networks (network without cycleGAN, residual squeeze and excitation attention gate network [rSEAG], ResUNet, and U-Net) hole and zoomed images in the in-focus plane (case_3). The all-display window range is 0–0.02. (**a**) Without cycleGAN, (**b**) rSEAG, (**c**) resUNet, and (**d**) U-Net.

**Figure 10 diagnostics-14-01957-f010:**
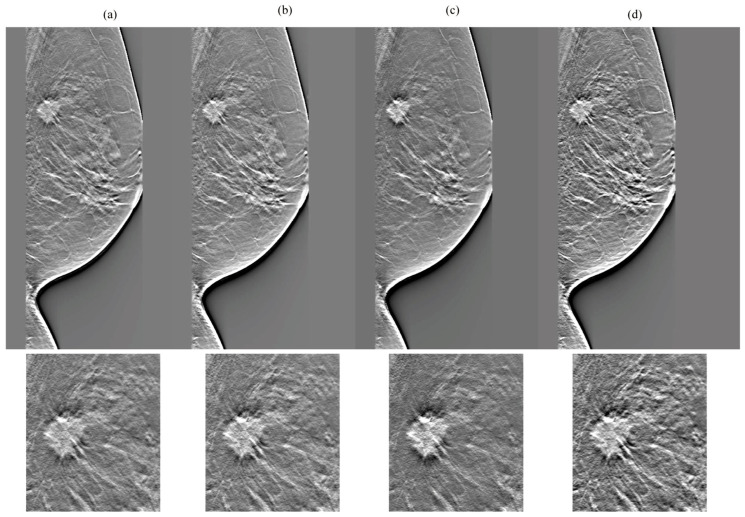
Comparisons between cycle-consistent generative adversarial network (cycleGAN) pre-reconstruction processing with and without relative generator networks (network without cycleGAN, residual squeeze and excitation attention gate network [rSEAG], ResUNet, and U-Net) hole and zoomed images in the in-focus plane (case_4). The all-display window range is 0–0.02. (**a**) Without cycleGAN, (**b**) rSEAG, (**c**) resUNet, and (**d**) U-Net.

**Figure 11 diagnostics-14-01957-f011:**
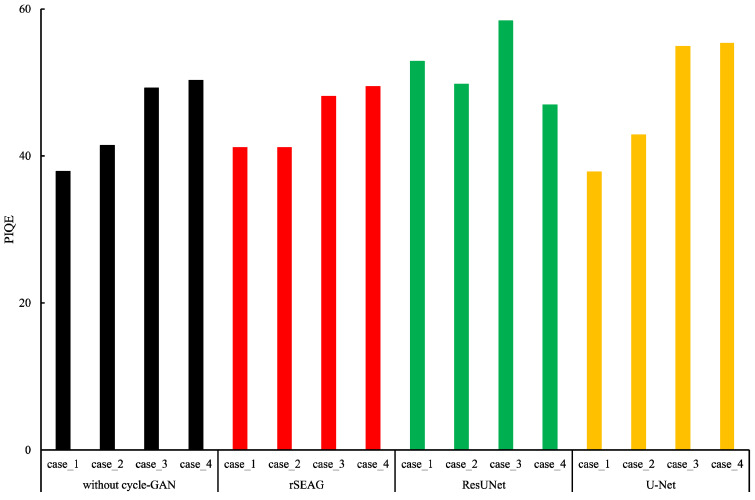
Results of the perception-based image quality evaluation (PIQE) versus each testing case. Plots of the PIQE for the in-focus plane at the cycle-consistent generative adversarial network (cycleGAN) pre-reconstruction processing with and without relative generator networks (network without cycleGAN, residual squeeze and excitation attention gate network [rSEAG], ResUNet, and U-Net) and each testing case. (Black: without cycle-GAN, Red: rSEAG, Green: ResUNet, Yellow: U-Net).

**Figure 12 diagnostics-14-01957-f012:**
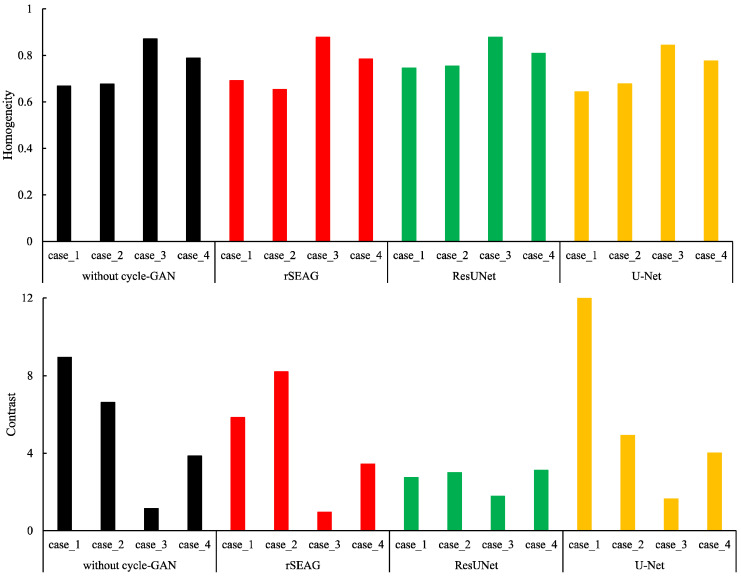
Results of the texture analysis (feature: “Homogeneity” and “Contrast”) versus each testing case. Plots of the “Homogeneity” and “Contrast” for the in-focus plane at the cycle-consistent generative adversarial network (cycleGAN) pre-reconstruction processing with and without relative generator networks (network without cycleGAN, residual squeeze and excitation attention gate network [rSEAG], ResUNet, and U-Net) and each testing case. (Black: without cycle-GAN, Red: rSEAG, Green: ResUNet, Yellow: U-Net).

**Figure 13 diagnostics-14-01957-f013:**
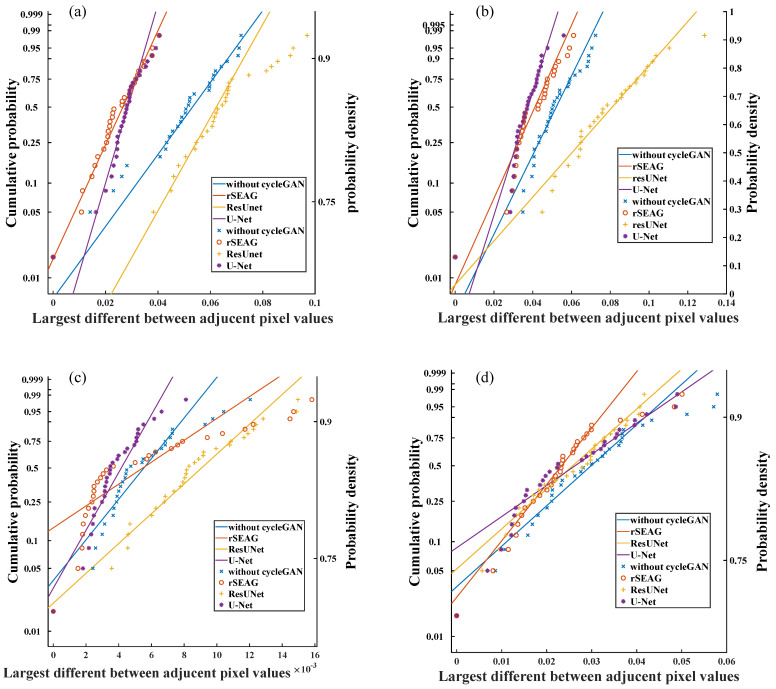
The largest variations extracted from 29 pixel−value profiles are plotted. The relatively large variations in pixel values were attributed to high-frequency ripple artifacts at different cases. (**a**) case_1, (**b**) case_2, (**c**) case_3, and (**d**) case_4.

**Table 1 diagnostics-14-01957-t001:** Clinical trial examination and corresponding datasets.

	Training Dataset (Reference)	Training Dataset (Object)	Testing Dataset
Mastopathy	7	5	-
DICS *	2	4	-
IDC **	10	26	4
Fibroadenoma	1	2	-
Findings without a diagnosis	25	8	-
Heterogeneous dense breasts			
(normal)	9	3	-
(benign or malignant)	-	3	2
Extremely dense breasts			
(normal)	4	-	-
(benign or malignant)	1	-	-
Average age ± SD *** (years)	58.46 ± 19.21	58.71 ± 13.54	49 ± 0.5
Acquisition view	CC ****	CC, MLO	CC, MLO
	MLO *****		
Number of cases	45	45	4
Number of total projection images	675	675	60
Number of total sample images	86,400	86,400	-
EI (average ± SD)	+219.8 ± 46.36	−495.42 ± 111.71	−570.5 ± 181.11
DI (average ± SD)	+5.68 ± 5.82	−12.75 ± 2.73	−12.5 ± 2.06

* Ductal carcinoma in situ; ** Invasive ductal carcinoma; *** Standard deviation; **** Cranio-caudal; ***** Mediolateral oblique.

**Table 2 diagnostics-14-01957-t002:** Summary of the cycleGAN hyperparameter settings in clinical trials.

Parameters	Values
Patch-per-image	128
Minibatch size	16
Iteration-per-epoch	5400
Patch size	128 × 128 pixels
Original projection image size	2560 × 4096 pixels
Crop projection image size	1024 × 2048 pixels
Generator architecture	rSEAG, ResUNet, U-Net
Discriminator architecture	PatchGAN
Initial learning rate	0.0001
Gradient decay factor	0.5
Squared gradient decay factor	0.999
Solver	Adam
Loss function	Adversarial loss
	Cycle consistency loss
	Fidelity loss (structural similarity: SSIM loss)

## Data Availability

All relevant data are within the manuscript and its nonpublished material files.

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
