# Peer review of "Pre-Reconstruction Processing with the Cycle-Consist Generative Adversarial Network Combined with Attention Gate to Improve Image Quality in Digital Breast Tomosynthesis"

_diagnostics, 2024, doi:10.3390/diagnostics14171957_

Round 1

Reviewer 1 Report

Comments and Suggestions for Authors

The study proposed and evaluated residual squeeze and excitation attention gate (rSEAG), a novel network that can improve image quality by reducing distortion attributed to artifacts and noise and by improving contrast. The paper is well written and achieved remarkable results. My decision is accept in present form.

The method proposed gave good results. U-net method seems having better results according to the visuals.

I have one question for the paper. What is the reason that authors only selected Homogeneity and Contrast features in GLCM method? Why were the other features not used for quality measurement?

Author Response

Please refer to the attached files for responses to reviewer's comments and corrections.

Reviewer 2 Report

Comments and Suggestions for Authors

Comments on the Quality of English Language

Good

Author Response

(The authors gave the same response as above.)
